# Bioaerosol Sampling Devices and Pretreatment for Bacterial Characterization: Theoretical Differences and a Field Experience in a Wastewater Treatment Plant

**DOI:** 10.3390/microorganisms12050965

**Published:** 2024-05-10

**Authors:** Anastasia Serena Gaetano, Sabrina Semeraro, Samuele Greco, Enrico Greco, Andrea Cain, Maria Grazia Perrone, Alberto Pallavicini, Sabina Licen, Stefano Fornasaro, Pierluigi Barbieri

**Affiliations:** 1Department of Chemical and Pharmaceutical Sciences, University of Trieste, Via L. Giorgieri, 1, 34127 Trieste, Italy; anastasiaserena.gaetano@phd.units.it (A.S.G.); ssemeraro@units.it (S.S.); enrico.greco@units.it (E.G.); slicen@units.it (S.L.); sfornasaro@units.it (S.F.); 2INSTM National Interuniversity Consortium of Materials Science and Technology, Via G. Giusti, 9, 50121 Firenze, Italy; 3Department of Life Sciences, University of Trieste, Via L. Giorgieri, 5, 34127 Trieste, Italy; pallavic@units.it; 4ACEGAS APS AMGA S.p.a., Via degli Alti Forni, 11, 34121 Trieste, Italy; andrea.cain@acegasapsamga.it; 5TCR Tecora, Via delle Primule, 16, 20815 Cogliate, MB, Italy; mariagrazia.perrone@tcrtecora.com

**Keywords:** bioaerosol, sampling, condensation growth tube, swirling aerosol sampler, gelatin filtration, wastewater treatment plant, metabarcoding, bacterial community

## Abstract

Studies on bioaerosol bacterial biodiversity have relevance in both ecological and health contexts, and molecular methods, such as 16S rRNA gene-based barcoded sequencing, provide efficient tools for the analysis of airborne bacterial communities. Standardized methods for sampling and analysis of bioaerosol DNA are lacking, thus hampering the comparison of results from studies implementing different devices and procedures. Three samplers that use gelatin filtration, swirling aerosol collection, and condensation growth tubes for collecting bioaerosol at an aeration tank of a wastewater treatment plant in Trieste (Italy) were used to determine the bacterial biodiversity. Wastewater samples were collected directly from the untreated sewage to obtain a true representation of the microbiological community present in the plant. Different samplers and collection media provide an indication of the different grades of biodiversity, with condensation growth tubes and DNA/RNA shield^TM^ capturing the richer bacterial genera. Overall, in terms of relative abundance, the air samples have a lower number of bacterial genera (64 OTUs) than the wastewater ones (75 OTUs). Using the metabarcoding approach to aerosol samples, we provide the first preliminary step toward the understanding of a significant diversity between different air sampling systems, enabling the scientific community to orient research towards the most informative sampling strategy.

## 1. Introduction

Bioaerosol is the aerosol containing components of biological origin that range in size from several nm to tens of μm [1]; it has relevant ecological [2,3] and health effects [4] when acting as a carrier of pathogens such as viruses, bacteria or allergens. The social impact of airborne pathogens is driving research and innovation to improve bioaerosol control and abatement technologies [5], including innovative photocatalytic systems and functional nanomaterials [6,7,8]. The air is a very complex matrix due to its dynamism, experiencing significant variation in the quantity, size, and composition of particles suspended within it. Given the variety of bioaerosols and the various ways in which they can interact, obtaining a representative sample is quite challenging. Additionally, the natural characteristic of the air matrix to dilute substances due to diffusion processes makes it difficult to capture specific microbiological events. The aim of the research heavily influences the selection of methods for collecting and analyzing samples. Different sampling devices can affect the results due to variations in their capability of separating particles by size, their efficiency, and the amount of sample they can collect. These differences can impact on the reliability of comparing results between studies, particularly if their objectives differ. Hence, there is a clear necessity for standardizing measurement procedures. Despite the wide array of research objectives, ensuring consistency within each approach is crucial for meaningful comparisons across different laboratories and experiments [9]. Culture-based approaches have been used traditionally to study bioaerosols; however, only a small portion of the overall microbial community is captured by culturing.

Molecular methods are thus applied for extending aerosol biodiversity characterization [10]. Currently, there are many different active bioaerosol sampling systems employed, with several others still in a development stage. Intercomparison studies have been published [11], considering eight different samplers and a commercial swirling aerosol sampler as a reference, in controlled laboratory situations. Among new devices, condensation growth tubes are promising ones, but they are still limitedly compared to other alternatives [12]. Some studies report on the performances of custom devices in both outdoor [13] and controlled settings [14]; other ones provide literature reviews without direct strict comparability in real settings [15]; and one study from a military research group provides qualitative indications on characteristics of devices [16]. An intercomparison was proposed considering sampling efficacy on aerosolized viruses [17] putting in strict proximity devices with highly different sampling flows, possibly generating interferences. A comparison between different sampling systems in the same outdoor environment can be helpful, and relevant bioaerosol sources provide significant scenarios to develop such experimental activities; wastewater treatment plants (WWTPs) [18] are considered among such sources, collecting microorganisms from excretions of inhabitants of a certain area and treating sewage inputs by degradation of organic matter producing a small but not negligible generation of bioaerosols.

The sewage service collects, lifts, removes, and purifies the wastewater from basements of individual residences, industrial sites, urban centers, roadways, and public spaces. The gathered water is transported to the purification plant, where it is treated and returned to the environment. In a complex matrix like wastewater, an extensive variety of chemical and biological substances can be found, such as pharmaceuticals, exogenous contaminants [19], nutrient concentrations, and microorganisms [20]. These biological substances have been studied for decades, but mostly to monitor substance concentrations entering the wastewater treatment process or to monitor removal efficiencies of the WWTPs. It is interesting to note that the presence of various substances in wastewaters gives direct qualitative and quantitative information about the behavior of inhabitants within a certain wastewater catchment. In particular, wastewater represents the whole enteric pathogen load from a local catchment region and captures community disease burden, therefore providing an ideal matrix for disease monitoring and surveillance of human activity [21]. The first step in the process of treating urban wastewaters is the removal of the larger solid bodies thanks to a system of progressively finer grids which trap coarse objects like pieces of plastic, small branches, and leaves. Wastewater next passes into aeration tanks, where degritting takes place: a flow of air is introduced into the waters to raise oils to the surface, while also precipitating sands and loam to the bottom. According to some studies, the intense mixing and turbulence occurring in the aeration tank results in splashing, bubble bursting, and spraying, all of which contribute to the release of bioaerosols [22,23]. The intensity of this phenomenon is such that it poses even a health risk for WWTP employees, who are more prone than the general population to have a wide range of work-related diseases, such as respiratory and gastrointestinal ones [24,25]. Recently, these peculiar characteristics of wastewaters have been used as a surveillance technique called Wastewater-Based Epidemiology (WBE) [21]. WBE can be performed in many different ways, also using molecular biology techniques. One of these can be the environmental DNA (eDNA) metabarcoding, which is a technique that allows for the simultaneous identification of many taxa in the same environmental sample thanks to the analysis of short DNA sequences of one or few genes called DNA barcodes [26,27,28]. This technique can be used for many different matrices (i.e., air and water) and organisms; and, in particular, for prokaryotes, the 16S rRNA gene is used as barcode. This particular region is chosen for its evolutionary conservation, which allows for the use of a combination of conserved universal Polymerase Chain Reaction (PCR) primers flanking a short hypervariable region (V3 and V4), which will be amplified and sequenced by high-throughput Next-Generation Sequencing (NGS) [29]. The output of this process allows us to identify and determine the diversity of microorganisms thanks to reference libraries, which are databases containing the DNA barcode assigned to previously identified taxa [30]. Overall, WBE based on eDNA metabarcoding is a non-invasive, efficient, cost-effective, and sensitive method which has the potential to supplement other approaches of biodiversity monitoring for ecological study and management on large spatial and temporal scales [31]. Even if the process of eDNA metabarcoding can produce accurately and inexpensively very large quantities of data, currently there is a lack of standardization and unification regarding bioaerosol sampling for molecular ecology research, mainly because the optimization process needed for each sampler use and the sample processing procedure significantly hinder the advance of bioaerosol science. Microorganisms in the air, unlike in other environmental matrices, such as water or soil, are present in the environment at low concentrations; therefore, it is crucial, first of all, to ensure that a sufficient quantity of genetic material is collected during the sampling phase in order to proceed with subsequent analyses. Additionally, the results of the research will also be influenced by the nucleic acid extraction protocol used, as it may have a greater or lesser recovery efficiency [32,33,34]. Due to these factors, it is difficult to compare different bioaerosol studies and, at the same time, understand how much any individual variable can influence the results [35]. The concentration, size distribution, and bacterial population of the bioaerosols that aerosolize from the wastewaters of WWTPs are not identical, and they depend on the different regions, the kind of wastewater treated, the season, the WWTP process selected, the technologies, and meteorological parameters [36,37,38,39]. In addition to the intrinsic factors of the WWTP, bioaerosol concentrations and compositions can be influenced by the equipment selection, the sampling time, and the adopted process [40]. Potential bacterial pathogens such as *Acinetobacter*, *Alcaligenes*, *Bacteroides*, *Chryseobacterium*, *Micrococcus*, *Enterobacter*, *Pantoea*, *Pseudomonas*, *Serratia*, and *Stenotrophomonas* have been detected in WWTP bioaerosols [41,42,43]. Other studies show WWTP bioaerosols affiliated with the phyla Proteobacteria, Bacteroidetes, and Firmicutes [40], while others identify heterotrophic and mesophilic bacteria belonging to genera *Pseudomonas*, *Micrococcus*, *Escherichia*, *Bacillus*, *Streptococcus*, *Staphylococcus*, *Klebsiella*, *Mycobacterium*, *Acinetobacter*, *Actinomyces*, and *Clostridium* [44]. In one study conducted in Italy, geographically matching the present work, the most abundant genera of bacteria were *Bacillus*, *Acinetobacter*, and *Arcobacter* [45]. A detailed report of the dominant microbial communities detected in the bioaerosol of the WWTP environment was recently published by Singh and colleagues [46]. However, most of the studies found in the literature are based on traditional cultural methods, which can surely lead to a higher quantity of biomass necessary for the subsequent analyses, on the one hand; but, on the other hand, they can lead to an underestimation of the microbiological richness and biodiversity of the sampled community. Recent studies on WWTP bioaerosols characterized by 16S rRNA gene metabarcoding are compared in Appendix A, showing how different choices in parameters for quantitation and experimental designs make results hardly comparable.

Following the 16S rRNA gene metabarcoding technique, we started with a characterization of the bacterial community present in the wastewaters of the WWTP of Trieste (Italy) to define a sampling strategy and laboratory protocols for the identification of the species present in the aeration tank of the WWTP, which is the aeration process expected to generate bioaerosol in significant quantities and on a constant basis [25,47]. Since there are no standards or guidelines which regulate outdoor bioaerosol sampling, with this study, we tested the performances and the sampling efficiency of three different commercial bioaerosol samplers that implement different aerosol collection principles and remarkable characteristics for the widespread use (swirling aerosol collection), ease of operability (gelatine filtration), and bioaerosol collection efficacy (condensation growth tubes). In particular, the present study focused also on the development of a procedure for the characterization of the microorganisms in the environmental bioaerosol by comparing three different sampling devices: the Airport MD8 (Sartorius, Goettingen, Germany), the BioSampler (SKC, Eighty Four, PA, USA), and the BioSpot-VIVAS (Aerosol Devices Inc., Ft. Collins, CO, USA).

## 2. Materials and Methods

### 2.1. Bioaerosol Samples

A total of 4 bioaerosol samples were collected as close as possible to the aeration tank emissions of the WWTP in Servola (Trieste, Italy) (i.e., 40 cm above the surface of the aeration tank water level, for which it must be added the height of each sampler that corresponds with the inlet of the devices) with three different sampling devices: Airport MD8 (sample ID: “MD8_tank”), BioSampler (sample ID: “SKC_PBS_tank”), BioSpot-VIVAS (samples ID: “Vivas_PS_tank”, and “Vivas_PBS_tank”). All samples were collected on the same day, starting collection simultaneously, except for “Vivas_PBS_tank”, which was added to obtain a hint on the relevance of type of bacterial collection medium (DNA/RNA shield™, as recommended by [12], vs. PBS, according to manufacturer’s recommendation) in aerosol sampling; DNA/RNA shield^TM^ could not be used with BioSampler due to generation of foam in the impinger, in addition to the costs related to a higher liquid collection volume for BioSampler (20 mL) vs. BioSpot-VIVAS (2.5 mL).

Since one of the aims of the study was to compare three different sampling devices for bioaerosol collection, an overall sampling volume of 1440 L was fixed for each sampler in accordance with the time of setting up, sampling, and availability of the structure. The instruments operated in field, being activated at the same time, with parameters set as summarized in Table 1.

#### 2.1.1. Airport MD8

The Airport MD8 is an air sampler used for the collection of microorganisms in indoor and outdoor environments. It is a sampling device that is easy to handle, portable, and quite small (300 mm × 135 mm × 165 mm for approximately 2.5 kg) and consists of a vacuum pump with timer and volumetric flow rate control. It can be used combined with gelatin membrane filters as an air filtration system, choosing the desired flow rate between four different values (10, 30, 40, and 50 L/min). This type of sampling system offers the possibility to solubilize the gelatin membrane filters in small liquid volumes (minimum 80–100 μL/cm^2^ filter area), allowing for further applications, such as microbial cultures and PCR [48] (see Table 2 for specifications). In the field, the gelatin filter was placed on the adapter of the sampler, using gloves and sterile tweezers. At the end of the sampling, the filter was placed in sterile Petri dishes sealed with parafilm and stored on ice until the arrival at the laboratory, where it was maintained at −20 °C. Before the DNA extraction, the filter was cut into pieces and placed in a clean tube, where sterile water was added in order to dissolve the gelatin. The tube was centrifugated for 10 min at 3000× *g* and then incubated at 37 °C for 10 min. The total DNA was isolated from the sample using the E.Z.N.A.^®^ Soil DNA Kit (Cat# D5625, Omega Bio-tek Inc., Norcross, GA, USA). A blank DNA extraction was performed as control to guarantee the absence of any environmental contamination. After the DNA extraction, the sample was concentrated using the DNA Clean & Concentrator^®^-5 Kit (Cat# D4013, Zymo Research Corporation, Irvine, CA, USA) in 8 μL of elution buffer. 

#### 2.1.2. BioSampler

The BioSampler is a liquid-based impinger widely used for bioaerosol sampling [49]. It is a lightweight glass sampler, quite small but fragile, that consists in an inlet, through which airborne particles pass into the collection device, and an outlet connected to a high-volume suction pump (mod. Bravo BIO, TCR TECORA^®^). In the collection vessel, three tangential nozzles ensure a swirling motion of the collection liquid upward on the inner walls. A detailed scheme of the device is provided in [50]. This peculiar design minimizes re-aerosolization and bounce of particles, preserving the integrity and viability of microorganisms like viruses, bacteria, fungi, and molds [50]. According to manufacturer’s recommendation, the BioSampler was operated at a flow rate of 12.5 L per minute (LPM), at which the cutoff size (50% efficiency) is about 300 nm [51] (see Table 2 for specifications). The collection liquid used was a phosphate-buffered saline solution (PBS), which was placed into the collection vessel for a total volume of 20 mL. Because of evaporation, at half of the sampling time, the collection vessel was refilled with PBS to 20 mL. Once the sampling finished, the PBS was transferred from the collection vessel into a fresh tube and stored on ice until the arrival at the laboratory. Here, the sample was concentrated by filtration on a 20 μm pore size cellulose filter, using a syringe. The filter was then placed in a sterile Petri dish, sealed with parafilm, and stored at −20 °C until the analyses. Before the extraction of the DNA, the filter was cut into little pieces and placed directly in the Distruptor Tube of the E.Z.N.A.^®^ Soil DNA Kit (Cat# D5625, Omega Bio-tek Inc., Norcross, GA, USA). A blank DNA extraction was performed as the control to guarantee the absence of any environmental contamination. After the extraction of the DNA, the sample was concentrated using the DNA Clean & Concentrator^®^-5 Kit (Cat# D4013, Zymo Research Corporation, Irvine, CA, USA) in 8 μL of elution buffer.

#### 2.1.3. BioSpot-VIVAS

The BioSpot-VIVAS is a bioaerosol sampler based on a condensation growth tube sampling method, which permits a gentle impingement for the wet collection of pre-concentrated air particles [52]. The flowrate of each growth tube is 1 L/min; therefore, the combined flowrate of all the eight growth tubes is 8 L/min, which is the total flowrate of the instrument in operation. A detailed scheme of the device is provided in [53]. Thanks to the overall low flowrate, which mimics the dynamics of particle deposition in human lungs, airborne particles are put under minimal stress when impacted onto the liquid surface of the collection solution, thus allowing for the sampling of viable microorganisms [53]. In particular, the BioSpot-VIVAS can sample a particle size range that varies from 5 nm to 10 μm, allowing for the sampling of ultrafine bioaerosol like viruses [54] (see Table 2 for specifications). The collection medium used was the DNA/RNA shield™, a preserving solution of nucleic acids, or PBS, which placed into a sterile 35 mm × 11 mm Petri dish for a total volume of 2.5 mL. After the sampling, the DNA/RNA shield™ was transferred to a fresh tube and stored on ice until the arrival at the laboratory, where it was stored at −20 °C (sample id: “Vivas_PBS_tank” and “Vivas_PS_tank”). The PBS was instead concentrated by filtration on a 20 μm pore size cellulose filter using a syringe. The filter was then placed in a sterile Petri dish, sealed with parafilm, and stored at −20 °C. The sample stored in DNA/RNA shield™ was directly processed with the E.Z.N.A.^®^ Soil DNA Kit (Cat# D5625, Omega Bio-tek Inc., Norcross, GA, USA), while the filtered sample was first cut into little pieces and then placed in the Distruptor Tube of the kit. After the extraction of the DNA, the sample stored in PBS was concentrated using the DNA Clean & Concentrator^®^-5 Kit (Cat# D4013, Zymo Research Corporation, Irvine, CA, USA) in 8 μL of elution buffer. A blank DNA extraction was performed as the control to guarantee the absence of any environmental contamination.

### 2.2. Wastewater

In order to have a true representation of the microbiological community present in wastewater, in addition to bioaerosol samples, 11 water samples of 15 mL of volume were collected directly from the sewage. In particular, seven samples were collected by grab sampling for each day (sample IDs: “Water_1”, “Water_2”, “Water_3”, “Water_4”, “Water_5”, “Water_6”, and “Water_7”), while the other four were the result of a composite process of sampling of 24 h in the same days, for which a small amount of water was sampled every hour over the course of the 24 h and stored in a single tank (sample IDs: “Water_24h_1”, “Water_24h_2”, “Water_24h_3”, and “Water_24h_4”). See Table 3 for the experimental design. Once in the laboratory, water samples were concentrated on a 20 μm pore size cellulose filter, using a syringe. The filter was then placed in a sterile Petri dish, sealed with parafilm, and stored at −20 °C until the analyses. For the extraction of the total DNA, the filter was first cut into little pieces and then placed directly in the Distruptor Tube, following the protocol of the E.Z.N.A.^®^ Soil DNA Kit (Cat# D5625, Omega Bio-tek Inc., Norcross, GA, USA). A blank DNA extraction was performed as the control to guarantee the absence of any environmental contamination.

### 2.3. qPCR, Library Preparation, and Sequencing

After the DNA isolation, a quantitative real-time PCR reaction was performed on a CFX96 Touch Real-Time PCR Detection System on a C1000 Touch Chassis (Bio-Rad, Hercules, CA, USA). The isolated DNA was used as a template to amplify the V3–V4 hypervariable region of the 16S rRNA gene, using three PCR primers, in order to reduce amplification biases: 515 Forward 5′-GTGYCAGCMGCCGCGGTAA-3′ [55], 806 Reverse 5′-GGACTACNVGGGTWTCTAAT-3′ [55], and 802 Reverse 5′-TACNVGGGTATCTAATCC-3′ [56]. PCR conditions were identical for all samples. Reactions were performed in a total volume of 15 μL reaction mix, composed of 2 μL of DNA template, 7.5 μL of AccuStart II PCR SuperMix (QuantaBio, Beverly, MA, USA), 0.6 μL of 515 Forward primer (10 μM), 0.3 μL of 802 Reverse primer (10 μM), 0.3 μL of 806 Reverse primer (10 μM), 0.75 μL of EvaGreen™ 20× (Biotium, Fremont, CA, USA) dye, and 3.55 μL of DNase-free water. The following thermal cycles were used: 94 °C for 3′, 94 °C for 20″, 55 °C for 30″, and 72 °C for 1′. The second PCR amplification was performed in a total reaction volume of 25 μL, containing the same reagents as the first one, but adding 1.5 μL barcoded/TrP1 (10 μM) as the primer and 1 μL of the first PCR amplification as the template. The following conditions were used: 94 °C for 3′, 94 °C for 10″, 60 °C for 10″, and 72 °C for 30″. A purification was performed for all the samples using the Mag-Bind^®^ Total Pure NGS kit (Omega Bio-tek Inc., Norcross, GA, USA), and each library was quality checked on agarose gel electrophoresis and quantified with Qubit™ Fluorometer (Thermo Fisher Scientific, Waltham, MA, USA). Finally, libraries were pooled in equimolar amounts and sequenced on an Ion Torrent PGM System.

### 2.4. Bioinformatic Analysis

The raw reads obtained within the frame of this study were imported into the CLC Genomics Workbench environment (Quiagen, Hilden, Germany). The quality of raw reads was evaluated with the Quality Control (QC) for sequencing reads tool to have a broad perspective of the sequencing process result and to optimize the subsequent trimming steps. The trimming operation was performed to increase the quality of the reads, and a QC of the trimmed reads was generated to evaluate the efficiency of the trimming process. On trimmed reads, an operational taxonomic unit (OTU) clustering was performed using the SILVA SSU 99% (v138.1) as reference database [57], with a similarity threshold of 97%. OTUs were then aggregated at genus level, and a filter selection was performed with the following parameters: “ID does not contain N/A”, “combined abundance ≥ 100”, and “taxonomy does not contain chloroplast”. The OTU table was exported, and the relative abundance of bacterial genera and the alpha diversity rarefaction curves were obtained and plotted within a Python environment.

## 3. Results

The sequencing process provided 129167 sequences (available at NCBI under the bioproject ID PRJNA1083383), with a median length of 250 bp. A total of 86 188 reads with an average length of 250 bp were kept after quality filtering, and their clustering resulted in 80 OTUs. A total of 70 087 trimmed reads and 75 OTUs were obtained from water samples, and 16 101 of trimmed reads and 64 OTUs were obtained from air samples (Appendix A). In the aerosol samples, the bacterial groups were all dominated by the following bacteria OTUs: *Stenotrophomonas* (64.24% of total sequences in all aerosol samples), *Klebsiella* (14.92%), *Enterobacter* (4.54%), *Acinetobacter* (3.13%), *Delftia* (2.85%), and *Pseudomonas* (2.62%). In the wastewater samples, the most abundant genera of bacteria were *Acinetobacter* (26.37% of total sequences in all water samples), *Arcobacter* (17.44%), Arcobacteraceae (13.7%), *Pseudarcobacter* (5.19%), *Bacteroides* (4.94%), and *Aeromonas* (4.92%).

There is clear evidence of a difference in bacterial biodiversity between aerosol and wastewater samples: in wastewaters, an unambiguous pattern with a comparable biodiversity is observed between all the samples collected, while for bioaerosol, a certain degree of genus-level abundance variability appeared in relation to the different sampler and mediums (Figure 1 and Appendix A). In particular, among the air samples, the “Vivas_PS_tank” contains a larger number of genera compared to the others, which instead displayed a striking dominance of bacteria of genus *Stenotrophomonas*. The most abundant genera found in this sample are *Klebsiella*, *Enterobacter*, *Pseudomonas*, and *Acinetobacter*. A significant difference can also be noticed between the two samples collected both by the BioSpot-VIVAS, but only differing for the collection medium (“Vivas_PBS_tank”, with the PBS as collection medium, and “Vivas_PS_tank”, with the DNA/RNA shield™ as collection medium). Also, in this case, the sample collected on the DNA/RNA shield™ medium shows a larger number of genera. Among the wastewater samples, the grab and the composite samples have comparable abundances of bacterial genera.

The difference in community compositions between air and wastewater samples is further supported by the alpha diversity rarefaction curves of the bacterial genus richness (Figure 2). The air samples show a smaller amount of reads and number of genera in respect of wastewater samples. The rarefaction curves demonstrated a good depth of coverage for wastewaters, with leveling of the curves by approximately 2000 reads. The same curves show differences between the different air samplers; in particular, “SKC_PBS_tank” and “Vivas_PS_tank” are those that reach the greatest number of sampled genera. Of the two, the “SKC_PBS_tank” reaches the highest number of genera but does not reach the plateau phase, failing to be representative of the bacterial community of our study. On the other hand, the “Vivas_PS_tank”, while reaching a lower absolute number of observed genera, shows an early plateau phase, suggesting that the BioSpot-VIVAS sampler is able to capture a more representative sample of the bacterial community. These results are confirmed by the relative abundance barplots (Figure 1).

Comparing all the samples collected at the aeration tank during the same day of both air (“MD8_tank”, “SKC_PBS_tank”, “Vivas_PS_tank”), and wastewater (“Water_24h_4”, “Water_6”), we determined that the total number of genera was 78. In particular, all the genera that were present in air samples (62 genera) were also present in wastewater samples, which, however, showed 16 more genera of bacteria.

## 4. Discussion

With this study, we provide an analysis of the bacterial community present in the aerosol and wastewater of the aeration tank of the WWTP of Trieste, using the 16S rRNA metabarcoding technique, providing also new insights about different air sampling system devices.

### 4.1. Bioaerosol Samples

Overall, the air samples show a lower number of genera than the wastewater ones in terms of relative abundance and alpha diversity rarefaction curves of the bacterial richness. Air samples are not as homogenous to each other as the water ones, and it seems that different sampling systems have different sampling efficiencies; therefore, the results obtained from a bioaerosol study may differ according to the air sampler used. In order to highlight and give support to the differences among the bioaerosol samples in the absence of replicates, we applied bootstrap and Euclidean pairwise distances to the community composition in terms of relative species abundances (Appendix A). Comparing air and wastewater samples of the same day, we found that the number of aerosolized genera of bacteria was lower than that present in the water of the aeration tank of the WWTP (62 vs. 78), but all the genera found in the air were derived presumably from wastewater. This can lead us to hypothesize that only a small part of the total number of bacteria is transferred from the liquid phase to the air, based on the ability of the microorganism to endure harsh physical stresses such as desiccation, UV exposure, or sampling stress. Many of the genera found in the air samples are composed of species that are capable of creating biofilms or that can even survive the treatments of a WWTP. These organisms can therefore, at the same time, be transported more easily from water to the air matrix and remain intact during the sampling phase. It is evident that microbial distribution patterns do not adhere to the widely accepted “everything is everywhere” theory, even though the data are still scarce. According to the findings, even if bacteria are microscopic, dispersal by air currents does not ensure that they reach every habitat equally, and arrival rates are low enough to be surpassed by local diversity due to natural selection. This can explain the fact that the air samples have a lower quantity of genera in respect to the grab wastewater samples.

We discuss below the six most abundant genera found in all the aerosol samples.

*Stenotrophomonas* is a genus of ubiquitarian bacteria, found mostly in soil, plants, and wastewaters [58,59]. It plays a significant ecological role in the nitrogen and sulfur cycles, and some species can provide benefits for plants, making them potential candidates for biotechnology uses in agriculture [60]. Nevertheless, a species called *S. maltophilia* is emerging as human pathogen causing fatal bacteremic infections and pneumonia [61]. It is known that *S. maltophilia* can endure in activated sludge of WWTPs, as well as chlorine or UV disinfection, because of its capacity to export toxic metabolites from the periplasm and to create biofilms, which serve as a protective barrier [62,63,64]. As a result, *S. maltophilia* can survive the WWTP processes; hence, it can be released into the aquatic environment, thus contributing to the spread of antibiotic-resistance determinants. *Klebsiella* is a genus of bacteria that can be found in a multitude of ecological niches, including soil, water, plants, birds, and mammals, both free-living and host-associated. In humans, it is typically present in the nose, throat, skin, and intestinal tract and can cause urinary tract and bloodstream infections, sepsis, or pneumonia [65]. Recently, several strains of this genus have posed a serious threat to clinical and public health on a global scale, as *Klebsiella* gained the ability to acquire gene and mutations for antibiotic resistance or for increasing its virulence features [66,67,68]. These opportunistic pathogens can be released into the environment mainly through sewage, and it is documented that they can survive even after the treatments of the WWTPs [69]. For instance, *Klebsiella pneumoniae* can survive or even develop virulence and antibiotic resistance in sewage environment; in fact, identical *K. pneumoniae* sequence types have been reported in both clinical settings and wastewater [70,71]. The genus *Enterobacter* belongs to a variety of environmental habitats, from soil to water and sewage, but its species can be also natural commensals of the animal and human gut microbiota [72]. *Enterobacter* species are part of the ESKAPE group of bacteria (*Enterococcus faecium*, *Staphylococcus aureus*, *Klebsiella pneumoniae*, *Acinetobacter baumannii*, *Pseudomonas aeruginosa*, and *Enterobacter* species), which are directly referred to resistant nosocomial infections [73,74]. Due to their adaptation to the hospital environment and their ease of acquiring several genetic mobile elements, including resistance and virulence genes, these bacteria are usually linked to multidrug resistance [75,76]. Different species of *Enterobacter* genus have been found in air samples from the WWTP environment, like potentially pathogenic *Escherichia coli* [77]. The *Acinetobacter* genus is ubiquitous in nature, being found in soil, water, or animals and humans [78]. It can be frequently found in WWTPs due to their capacity to create biofilms on both biotic and abiotic surfaces, being therefore resistant to different environmental factors, such as desiccation and disinfectants [79,80]. Due to its resistance to environmental factors, it can be found also in the air matrix [81] and was found in the WWTP bioaerosol by Zhang and colleagues [81]. It is supposed that, in activated sludge, *Acinetobacter* spp. experience a change in metabolism, which may be a strategy used to survive and persist to the wastewater treatment process [82]. Many *Acinetobacter* species include opportunistic pathogens and are among the most frequent causes of hospital infections, and, in particular, they are now gaining interest because of their ability to acquire antibiotic-resistance genes [83,84,85,86]. *Delftia* strains have been isolated from different environments, including contaminated soil [87], wastewaters [88], WWTP bioaerosols [89], and hospitals [90]. The members of this genus are plant growth-promoting bacteria [91] and can transform or degrade several organic pollutants [92,93]. There is evidence on the possibility that metal contamination of natural environments may play a significant role in the spread of microbial antibiotic resistance; in fact, it is known that some *Delftia* sp. isolates may be resistant to a variety of antibiotics [94]. Members of the *Pseudomonas* genus are considered to be one of the most varied and widespread groups of bacteria, found in a variety of natural, clinical, and artificial environments [95], as well as in WWTPs [96]. A key characteristic of *Pseudomonas* species is the ability to produce extracellular products like polymeric substances, which have been implicated in attachment processes, biofilm formation, and virulence [97]. Members of the *Pseudomonas* genus are thought to be able to acquire nearly all known antimicrobial-resistance mechanisms thanks to their genome plasticity [98]. Some are able to adopt resistant forms, like spores, to survive in hostile environments; thus, dormant organisms can be released into the environment, along with any resistance genes they may have acquired, for example, in hospital settings [99].

### 4.2. Wastewater Samples

The results are consistent with previous wastewater studies, which found the genera *Acinetobacter* and *Arcobacter* as dominant and common members of WWTPs worldwide [100,101,102,103,104,105]. The *Acinetobacter* genus includes Gram-negative aerobic species generally found in soil, water, sewage, and also clinical environments. In fact, the genus increased interest in the health-care field for the emergence of multiresistant strains, some of which are pan-resistant to antibiotics [106]. *Acinetobacter baumannii* is one of the most prevalent causes of nosocomial infections [107], and it is considered the World Health Organization’s number one critical priority pathogen for which new therapeutics are urgently required [108]. The genus *Arcobacter* has become increasingly important in medical, veterinary, and food safety fields because of its emergent enteropathogenesis and potential zoonotic agents [109], as well as for the increased prevalence of antibiotic-resistance strains [110]. In fact, some *Arcobacter* species may cause various diseases in animals, such as reproductive problems, mastitis, and gastric ulcers; and in humans, including gastroenteritis, bacteremia, peritonitis, and endocarditis [111]. The same description can be ascribed to the genus *Pseudarcobacter*, later synonym of *Arcobacter* [112]. The family of *Arcobacteraceae* can be found in a variety of habitats, mainly in aquatic environments like groundwater and sewage, but also in food and food-processing facilities [111,113,114,115]. It hosts both animals and humans, being some species linked with intestinal diseases, bacteremia, and peritonitis [116]. *Bacteroides* are the most predominant anaerobe bacteria in mammals gut flora, playing a main commensal role in the microbial food webs by processing complex molecules [117]. Although this genus provides valuable insights into bacteria in maintaining health, some species are also linked to significant human diseases being opportunistic pathogens in other body locations [118,119]. *Aeromonas* is one of the most common genera found in wastewater microbial community [120], and it can also colonize soil, freshwater systems, or aquatic animals like fishes [121,122]. The presence of this genus in surface waters is linked to a high concentration of total and assimilable organic carbon, as certain *Aeromonas* species are reliable indicators of water quality and pollution. It is well known that some members of this genus acquired antibiotic resistance genes, and several studies have shown that *Aeromonas* may have the ability to generate and spread antibiotic resistance in various types of water bodies [123].

### 4.3. Comparison between Air Sampling System Devices

For what concerns bioaerosol collection in this field experience, the activation of the three samplers and operational control for allowing the sampling of 1440 L of air, MD8 with a light and compact body provided ease of collection of microorganisms on filters in the field, treating and extracting the gelatine support in laboratory. The BioSampler has a glass sampling body connected by tubes to an external suction pump, which operates optimally by generating turbulence and transferring bioaerosol from the air flow to collection liquid when the PBS volume is 20 mL; the volume should be checked and eventually refilled. The BioSpot-VIVAS is compact but comparatively heavy, and the positioning of the Petri dish filled with collection liquid and the relative cooling system resulted in not being comfortable for handling at the WWTP aeration tank.

According to data from this preliminary study, the liquid medium appears to be the most effective method for bioaerosol sampling, being the BioSpot-VIVAS and the BioSampler more efficient in respect of the MD8 Airport. In particular the DNA/RNA shield™ as the medium outperforms the PBS; in fact, it was not even necessary to proceed with the purification and concentration of the samples using the DNA Clean & Concentrator^®^-5 Kit after the DNA extraction. The DNA/RNA shield™ medium was only used for the BioSpot-VIVAS sampler since; in addition to the costs related to a higher liquid collection volume for the BioSampler (20 mL) vs. the BioSpot-VIVAS (2.5 mL), it is a solution that, when shaken, forms foam, and so it is incompatible with the swirling motion of the BioSampler device. These first results suggest that the BioSpot-VIVAS sampling system in combination with the DNA/RNA shield™ medium is the best method for bioaerosol sampling, as the BioSpot-VIVAS sample collected in the PBS medium showed poor biodiversity. The BioSpot-VIVAS is the sampler that operates at a lower flow rate than the others (8 LPM), and it is known that this characteristic allows for a physical collection efficiency above 95% for particles from 8 nm to 10 mm [54]. Thanks to a low water vapor condensation strategy, small particles are enlarged, and enlarged aerosols are collected more efficiently, enabling the resistance of desiccation and maintenance of viability during sampling. For what concerns the BioSampler, even if in a smaller number of reads, it succeeded to express a high biodiversity, slightly more than the BioSpot-VIVAS sample stored in DNA/RNA shield™ (see Figure 2). The low genetic abundance of the sample may be due both to the chosen medium (i.e., PBS), which is not preservative of the nucleic acids as the DNA/RNA shield™ is, and to the fact that the sampling volume of this device is 20 mL, in which the collected material is highly diluted compared to the 2.5 mL volume of the BioSpot-VIVAS sampler. In terms of operational characteristics for use in the outdoor environment, the Airport MD8 was the easiest sampler to use and clean. In fact, its dimensions and weight are smaller than that of the BioSpot-VIVAS, it is less fragile in respect of the BioSampler which has delicate nozzles, and it is entirely made of glass. The transportation on field of the BioSpot-VIVAS device is not easy in comparison with the other two instruments due to its weight (22.5 kg) and its dimensions (760 mm × 485 mm × 370 mm). The Airport MD8 does not require auxiliary equipment and has a reasonable duration of battery power; on contrary, the BioSampler needs to be connected to a high-suction pump, and both the BioSpot-VIVAS and the BioSampler systems need to be connected to an electrical power source.

## 5. Conclusions

Currently, no standardized methods for bioaerosol sampling exist, so using the metabarcoding approach to aerosol samples, the main implication of this study is that the present findings represent a first field experimental step toward the understanding of a significant diversity between different air sampling systems (i.e., filtration, swirling aerosol collection, and condensation growth tube), even if the sample size is critical to estimate a robust statistical power. Future directions in this sense could involve increasing the number of sampling days and, consequently, the number of samples for each device, given the operational availability in the plant.

In this screening comparison on bioaerosol samplers, we identified a candidate device and collection medium for the bioaerosol sampling, suggesting that BioSpot-VIVAS sampler with DNA/RNA Shield™ as the collection liquid is the best choice, despite lacking an optimal ease of operation due to the dimension, weight, and need for parameter tuning. Our data also highlight a difference in bacterial biodiversity between wastewater and air samples, with the latter being different depending on the device used based on the parameters set in this study (an overall sampling volume of 1440 L fixed for each sampler). All the genera found both in the wastewater and in the bioaerosol are consistent with other studies reported in the literature. However, since most of the studies reported in the literature are based on traditional cultural methods, which may result in an underestimation of the microbiological richness, it is difficult to make a true comparison with the abundances of the bioaerosol samples. Furthermore, the majority of the bioaerosol studies in the WWTP environment do not use the same sampling devices employed in the present work.

Focusing on possible human pathogens, the metabarcoding technique, combined with an automated sampling methodology, can be modified to validate and improve epidemiological models, as well as to organize early warning systems in biomonitoring approaches. To implement the study of the biodiversity present both in air and wastewater a metagenomic survey can be followed to identify a vast number of genomes and perform a functional characterization present in the samples. In addition, in-depth physiochemical and microbial characterization of bioaerosol will be considered in future studies in parallel with the sequencing approach. By taking into consideration smaller aerosol particles (i.e., <0.3 μm), effectively collected by BioSpot-VIVAS, the viral component of the bioaerosol could be investigated as well.

## Figures and Tables

**Figure 1 microorganisms-12-00965-f001:**
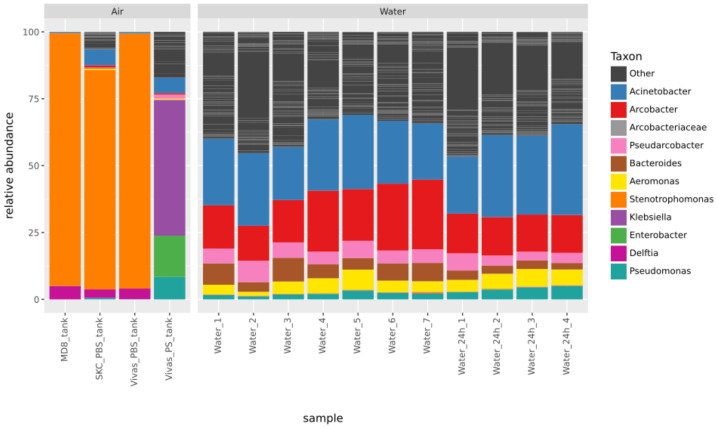
Relative abundance barplots of most dominant OTUs detected in air and wastewaters samples of the aeration tank of the WWTP of Servola (Trieste). Genera of the six most abundant OTUs in air and wastewater are color coded, and dark gray codes are for all genera that were grouped under the “Other” category. The air sample collected using the Airport MD8 has the sample ID “MD8_tank”. The air sample collected using the BioSampler has the sample ID “SKC_PBS_tank”. The air samples collected using the BioSpot-VIVAS have the sample IDs “Vivas_PBS_tank” (sampled using the PBS as collection media) and “Vivas_PS_tank” (sampled using the preserving solution of nucleic acids DNA/RNA shield™ as collection media). The wastewater samples that were collected by grab sampling for each day have the sample IDs “Water_1”, “Water_2”, “Water_3”, “Water_4”, “Water_5”, “Water_6”, and “Water_7”. The wastewater samples that were the result of a composite process of sampling of 24 h in the same days have the sample IDs “Water_24h_1”, “Water_24h_2”, “Water_24h_3”, and “Water_24h_4”.

**Figure 2 microorganisms-12-00965-f002:**
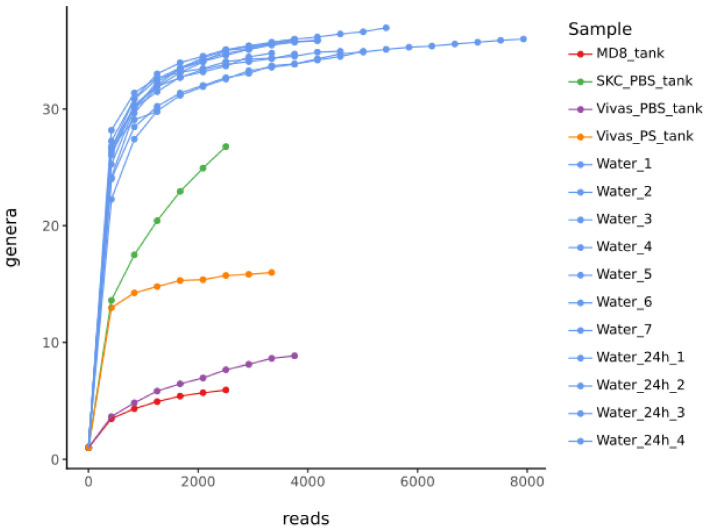
Alpha diversity rarefaction curves of bacterial communities based on the 16S rRNA gene sequences from air and wastewater sampled in the WWTP of Servola (Trieste). The x-axis represents the sequencing depth in number of reads, and the y-axis the estimation of the OTU richness detected at genus level. The air samples collected using the Airport MD8 has the sample ID “MD8_tank”. The air sample collected using the BioSampler has the sample ID “SKC_PBS_tank”. The air samples collected using the BioSpot-VIVAS have the sample IDs “Vivas_PBS_tank” (sampled using the PBS as collection media) and “Vivas_PS_tank” (sampled using the preserving solution of nucleic acids DNA/RNA shield™ as collection media). The wastewater samples that were collected by grab sampling for each day have the sample IDs “Water_1”, “Water_2”, “Water_3”, “Water_4”, “Water_5”, “Water_6”, and “Water_7”. The wastewater samples that were the result of a composite process of sampling of 24 h in the same days have the sample IDs “Water_24h_1”, “Water_24h_2”, “Water_24h_3”, and “Water_24h_4”.

**Table 1 microorganisms-12-00965-t001:** Summary of the experimental settings for each air sampling device.

Parameter	Airport MD8	BioSampler	BioSpot-VIVAS
Flow rate (L/min)	50	12.5	8
Total volume (L)	1440	1440	1440
Sampling Time	28.8 min	1.92 h	3 h

**Table 2 microorganisms-12-00965-t002:** Technical characteristics of the three air sampling devices: Airport MD8, BioSampler, and BioSpot-VIVAS.

Sampler Characteristics	Airport MD8	BioSampler	BioSpot-VIVAS
Supplier	Sartorius (Goettingen, Germany)	SKC Inc. (Eighty Four, PA, USA)	Aerosol Devices Inc., Ft. Collins, CO, USA
Sampling flow rate (L/min)	50	12.5	8
Sampling principle	Filtration	Swirling aerosol collection	Condensation growth tube
Support	Gelatine	Liquid	Liquid
Collection media	Gelatine	PBS	PBS, DNA/RNA shield™
Sampling volume	80 mm diameter	20 mL	2.5 mL
Weight	2.5 kg	0.16 kg *	24 kg
Dimensions	300 mm × 135 mm × 165 mm	220 mm × 50 mm × 50 mm *	760 mm × 485 mm × 370 mm

* Not including the high-volume suction pump needed for sampling.

**Table 3 microorganisms-12-00965-t003:** Experimental design of the samples collected during the study.

Wastewater	Sample ID	Sampling		Day
	Water_24h_1	Composite		2
	Water_24h_2	Composite		3
	Water_24h_3	Composite		5
	Water_24h_4	Composite		6
	Water_1	Grab		1
	Water_2	Grab		2
	Water_3	Grab		3
	Water_4	Grab		4
	Water_5	Grab		5
	Water_6	Grab		6
	Water_7	Grab		7
**Aerosol**	**Sample ID**	**Sampler**	**Support medium**	**Day**
	MD8_tank	Airport MD8	Gelatine filter	6
	SKC_PBS_tank	BioSampler	PBS	6
	Vivas_PS_tank	BioSpot-VIVAS	DNA/RNA shield™	6
	Vivas_PBS_tank	BioSpot-VIVAS	PBS	8

## Data Availability

All raw reads used for this work are available in public databases under the BioProject ID PRJNA1083383 on SRA (NCBI).

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
