# Peer review of "Bioaerosol Sampling Devices and Pretreatment for Bacterial Characterization: Theoretical Differences and a Field Experience in a Wastewater Treatment Plant"

_microorganisms, 2024, doi:10.3390/microorganisms12050965_

Round 1

Reviewer 1 Report

Comments and Suggestions for Authors

The work entitled “Bioaerosol sampling devices and pretreatment for bacterial characterization: theoretical differences and a field experience in a wastewater treatment plant”, authors are Anastasia Serena Gaetano, Sabrina Semeraro, Enrico Greco, Andrea Cain, Maria Grazia Perrone, Alberto Pallavicini, Samuele Greco, Sabina Licen, Stefano Fornasaro and Pierluigi Barbieri, is devoted to an interesting, mysterious and important topic of origin and diversity of microorganisms in the air. Bacterial diversity of aerosols formed near a wastewater treating plant was compared with that of wastewater itself. Some microbial groups, dangerous for workers, such as Enterobacter, Klebsiella and Pseudomonas were found in the air close to WWTP. Bacterial diversity was not strong but correlated with diversity in wastewater. Other interesting results were dependence of revealed aerosol biodiversity on the method of cell concentration and isolation and fact that some microbial groups dominated in wastewater samples were not presented in the air samples.

However, the experimental part has some gaps in the area of statistics (not enough information about replicates, controls and normalization) that is specified in comments.

Specific comments

1) It is strongly recommended to add number of replicates for each trial (MD8_ctrl, MD8-tank, SKC-PBS-tank, Vivas_PBS-tank, Vivas_PS_tank, Water_1, Water_2, Water_3, Water_4, Water_5, Water_6, Water_7, Water_24h_1, Water_24h_2, Water_24h_3, and Water_24h_4). Three replicates seem to be enough for a minimal appropriate statistical analysis. Current results without indication of the number of replicates generate doubts that after next sampling the same diversity will be found.

2) It is recommended to include controls (aerosols sampled far from the aerotank) for each trial (SKC-PBS-tank, Vivas_PBS-tank, and Vivas_PS_tank), not only for MD8. Another way is to exclude control variant for MD8 from the paper. If extra scientific task is to show changes in microbial gradients as a function of distance from the aerotank and how aerotank-borne microorganisms are distributed, it is better to perform control measurements. If the task is only to compare wastewater and aerosol diversity near the WWTP, it is better to not include control for MD8.

3) It is recommended to add distances for sampling close and far (control) from the aerotank.

4) How the sample volume normalization was performed? Was the volume of pumped air same for each device (MD8, BioSampler, and Vivas)? Diversities revealed with metagenomics can depend on the number of cells for DNA isolation, and rare groups can be underestimated if a small volume of air is analyzed.

Overall recommendation

Reconsider after major revision (control missing in some experiments)

Author Response

We would like to thank the reviewer for the precious advice, here follow our responses to the points raised:

1) It is strongly recommended to add number of replicates for each trial (MD8_ctrl, MD8-tank, SKC-PBS-tank, Vivas_PBS-tank, Vivas_PS_tank, Water_1, Water_2, Water_3, Water_4, Water_5, Water_6, Water_7, Water_24h_1, Water_24h_2, Water_24h_3, and Water_24h_4). Three replicates seem to be enough for a minimal appropriate statistical analysis. Current results without indication of the number of replicates generate doubts that after next sampling the same diversity will be found. 

Re: A table (Table 3) is now provided in order to clarify the sampling design and the characteristics of each sample. Surely, three replicates are the minimum amount for proper statistical analysis and this study lacks an appropriate sample size, which is critical to achieve statistical power. However, with this study we want to highlight, above all, the methods we used in the sampling and laboratory analysis phases. We demonstrated that a sufficient amount of genetic material was sampled for the subsequent genetic analyses, which is not obvious to reach when speaking of bioaerosol studies. In fact, most of the studies are based on culturable techniques that enable to enhance the biomass of the sample. As we stated in the paper at line 121, “Microorganisms in air, unlike in other environmental matrices such as water or soil, are present in the environment at low concentrations, therefore it is crucial first of all to ensure that a sufficient quantity of genetic material is collected during the sampling phase in order to proceed with subsequent analyses. Additionally, the results of the research will also be influenced by the nucleic acid extraction protocol used, as it may have a greater or lesser recovery efficiency”.

2) It is recommended to include controls (aerosols sampled far from the aerotank) for each trial (SKC-PBS-tank, Vivas_PBS-tank, and Vivas_PS_tank), not only for MD8. Another way is to exclude control variant for MD8 from the paper. If extra scientific task is to show changes in microbial gradients as a function of distance from the aerotank and how aerotank-borne microorganisms are distributed, it is better to perform control measurements. If the task is only to compare wastewater and aerosol diversity near the WWTP, it is better to not include control for MD8.

Re: The “MD8_CTRL” sample is now excluded from the paper. This sample was included because of the short sampling time of the Airport MD8 device, i.e., 30 minutes to reach the fixed total volume of 1440 L (the Airport MD8 device samples at 50 L/min). For the other sampling devices, it was not possible to perform environmental controls for logistical reasons as the sampling time was way larger to reach the fixed total volume of 1440 L (the BioSampler samples at 12.5 L/min and the BioSpot-VIVAS samples at 8 L/min).

3) It is recommended to add distances for sampling close and far (control) from the aerotank.

Re: a detailed distance from the aeration tank is now added in the text (line 140)

4) How the sample volume normalization was performed? Was the volume of pumped air same for each device (MD8, BioSampler, and Vivas)? Diversities revealed with metagenomics can depend on the number of cells for DNA isolation, and rare groups can be underestimated if a small volume of air is analyzed.

Re: The sample volume was fixed for each sampler for a total of 1440 L (as written in line 179), so each device sampled the same volume of air. This value was set in accordance with the maximum sampling time available in the plant for the sampler having the lowest flow rate (i.e., the BioSpot-VIVAS, 8 L/min) based on logistical reasons such as the working time of the plant and the setting time before and after the sampling. We tried to do our best with the time slot granted by the plant.

Reviewer 2 Report

Comments and Suggestions for Authors

Comments:

1.      Abstract: Since it is a research-based study, authors are suggested to include the main quantitative data.

2.      At the end of the abstract, add a brief statement on the important implications/contributions of this work.

3.      Line 20: Change “molecular methods, as 16S rRNA gene-based barcoded sequencing” to “molecular methods, such as 16S rRNA gene-based barcoded sequencing”

4.      Introduction: Overall, it mainly contains qualitative statements. There are many publications on the microbial characterization of bioaerosols collected from indoor and outdoor environments including from wastewater treatment plants (WWTPs). Thus, authors need to provide a paragraph comparative discussion on the quantitative information on the abundance and diversity of microbial communities found in WWTPs-based bioaerosols.

5.      At the last paragraph of the introduction, authors are required to state the key hypotheses that were tested in this work.

6.      Line 142: “A total of 5 bioaerosol samples were collected as close as possible to the aeration tank…” Are they collected from the same day? What is the shorted distance, a few centimeters above the surface of the aeration tank water level?

7.      Table 1: Give heading for column 1, may be “Parameter”.

8.      Table 2: Also, add a heading for column 1, may be “Sampler characteristics”.

9.      Figure 1: The Y-axis title should be changed from “abundance” to “Relative abundance”.

10.  Figure 1 and Figure 2: The figure legend can be expanded by explaining the abbreviations or terms used for the three samplers which were employed for bioaerosols collection.

11.  For better understanding to readers, authors are suggested to prepare a comparative table by summarizing the abundance of dominant microbial communities detected in the bioaerosols from this work with other recent WWTPs-based published studies on microbial aerosols. 

12.  Statistical analysis: It would be better to do some appropriate statistical analysis to show whether the variations of microbial composition obtained using three different samplers were statistically significant.

13.  At the end of discussion, develop a new section highlighting the major implications and future directions/limitation of the present work.

14.  In this study, only microbial characterization was done using the sequencing approach. However, authors are advised to consider doing in-depth physicochemical and microbial characterization of bioaerosols in future studies.

Comments on the Quality of English Language

Minor typographical and grammatical errors checking are required. 

Author Response

We would like to thank the reviewer for the precious advice, here follow our responses to the points raised:

  1. Abstract: Since it is a research-based study, authors are suggested to include the main quantitative data.

Re: The number of OTUs is now added at the end of the abstract (line 30-31)

  1. At the end of the abstract, add a brief statement on the important implications/contributions of this work.

Re: This is now added at the end of the abstract (line 31-34).

  1. Line 20: Change “molecular methods, as 16S rRNA gene-based barcoded sequencing” to “molecular methods, such as 16S rRNA gene-based barcoded sequencing”

Re: The sentence is now modified.

  1. Introduction: Overall, it mainly contains qualitative statements. There are many publications on the microbial characterization of bioaerosols collected from indoor and outdoor environments including from wastewater treatment plants (WWTPs). Thus, authors need to provide a paragraph comparative discussion on the quantitative information on the abundance and diversity of microbial communities found in WWTPs-based bioaerosols.

Re: A paragraph citing other bioaerosol studies conducted in wastewater treatment plants is now added (line 128-150), however a quantitative comparison between different studies is difficult to make since the majority of the bioaerosol studies in the WWTP environment do not use the same sampling devices employed in the present work. Furthermore, the laboratory techniques for the analysis of the samples are almost always based on traditional cultural methods (sometimes supported by the metabarcoding of the 16S gene technique - that was used for our study), which can lead to an underestimation of the microbiological richness. It is for these reasons that, as we stated in the paper, the research on bioaerosol would greatly benefit from standardization to achieve comparability between studies. Such standardization is lacking as of today. A comparative table of other WWTPs-based published studies on microbial aerosols in now added in the Supplementary Information as Supplementary table 1 (Table SI1), focusing on studies on non-culturable and culturable bacteria characterization by 16S rRNA gene metabarcoding, on bioaerosol from WWTPs: quantitation is provided by heterogeneous parameters (as CFU/m3 for culturable bacteria, ASV or OTU per sample, gene copies per ng of DNA) not allowing direct comparison among studies.  Qualitative analysis need to be contestualized: in general, predominant genera are identified, but studies differs in experimental designs, covering single or several seasons, looking for ratios among culturable and non-culturable bacteria, or focusing on pathogens. Pieces of information result hardly comparable.

  1. At the last paragraph of the introduction, authors are required to state the key hypotheses that were tested in this work.

Re: The hypothesis is now added.

  1. Line 142: “A total of 5 bioaerosol samples were collected as close as possible to the aeration tank…” Are they collected from the same day? What is the shorted distance, a few centimeters above the surface of the aeration tank water level?

Re: A table (Table 3) is now provided in order to clarify the sampling design and the characteristics of each sample. One control bioaerosol sample has been discarded following reviewer 1. The distance above the surface of the aeration tank water level is added as well (line 170).

  1. Table 1: Give heading for column 1, may be “Parameter”.

Re: Heading for column 1 (Table 1) is now added, but we believe that this space should be left blank because it is a double entry table.

  1. Table 2: Also, add a heading for column 1, may be “Sampler characteristics”.

Re: Same as the previous point, we consider this space should be blank because it is a double entry table but the heading is now added as requested.

  1. Figure 1: The Y-axis title should be changed from “abundance” to “Relative abundance”.

Re: The figure is now modified adding “Relative”

  1. Figure 1 and Figure 2: The figure legend can be expanded by explaining the abbreviations or terms used for the three samplers which were employed for bioaerosols collection. 

Re: We added in the caption the explanation of the samples shown in Figure 1 and Figure 2.

  1. For better understanding to readers, authors are suggested to prepare a comparative table by summarizing the abundance of dominant microbial communities detected in the bioaerosols from this work with other recent WWTPs-based published studies on microbial aerosols.  

Re: See Response of point number 3. Bibliography is citied in the “Discussion” section highlighting that the most abundant genera detected in the bioaerosol are in line with other studies conducted in WWTPs, but a quantitative comparison between the studies cannot be done because of different study designs.
A brief list of studies on the abundances of dominant microbial communities detected in the bioaerosol in WWTPs is reported in the “Introduction” section as requested in the point number 3. A comparative table of other WWTPs-based published studies on microbial aerosols in now added in the Supplementary Information as Supplementary table 1 (Table SI1).

  1. Statistical analysis: It would be better to do some appropriate statistical analysis to show whether the variations of microbial composition obtained using three different samplers were statistically significant. 

Re: An analysis of the differences between samples via bootstrap (n = 10000) distribution of mean and relative abundance and upper quartile of the same value was performed. A dendrogram relative to hierarchical clustering of relative abundance based on complete linkage of Euclidean pairwise distances was performed as well. The figures relative to these statistics are added to the Supplementary Information, as Supplementary Figure 1 a, b and c.
Due to the scarce sampling units, statistics like the p-value cannot be performed.

  1. At the end of discussion, develop a new section highlighting the major implications and future directions/limitation of the present work. 

Re: The section is now added and expanded in the “Conclusion” section.

  1. In this study, only microbial characterization was done using the sequencing approach. However, authors are advised to consider doing in-depth physicochemical and microbial characterization of bioaerosols in future studies.

Re: Thank you for your suggestions, these kind of analyses are added in the “Conclusion” section when introducing of the possible implementations of our study (line 584).

Reviewer 3 Report

Comments and Suggestions for Authors

The following improvements are required:

1. Improve description of aerosol sampling devices – add schemes of this equipment.

2. Clarify the description of the samples (aerosol and waste water):

Aerosol samples were collected using 3 devices and designated as follows:

·         Airport MD8: samples MD8_CTRL and MD8_tank;

·         Biosampler: samples SKC_PBS_tank;

·         BioSpot-VIVA: samples Vivas_PBS_tank and Vivas_PS_tank.

Clarify the difference between different samples collected using the same device (conditions, sampling site, etc.). On which day were these aerosol samples collected (day 1…..7)?

Clarify the correspondence of aerosol samples and water samples (collection day).

Water samples were designated “Water_1”, “Water_2”, “Water_3”, “Water_4”, “Water_5”, “Water_6”, and “Water_7”, “Water_24h_1”, “Water_24h_2”, “Water_24h_3”, and Water_24h_4”.

Clarify designations “Water_24h_1….4”. On which day were these samples collected (day 1…..7)?

3. Line 282. A total of 70 087 trimmed reads and 75 OTUs were obtained from water samples, and 18 598 of trimmed reads and 68 OTUs. Provide the data on the number of reads used for each sample (in the text / or the table).

4. Figure 1 and results description. The results for the sample “Vivas_PS_tank” significantly differ from those obtained for other aerosol samples. Describe sample “Vivas_PS_tank” and its differences from other aerosol samples.

5. Figure 1 shows only main genera revealed in the samples. Add the table (for example, in supplementary material) with the list of genera detected in the samples.

6. The Discussion provides the analysis of the results obtained in the present work and comparison of the devices used.

Compare the results obtained in your work with those obtained in other works to clarify novelty and significance of your work.

Author Response

We would like to thank the reviewer for the precious advice, here follow our responses to the points raised:

  1. Improve description of aerosol sampling devices – add schemes of this equipment.

Re: We citied in the text the bibliography containing the schemes of the devices used (line 217 and 241). However, for the simple (gelatin filter – pump) Airport MD8, no scheme was found online, but the commercial brochure (https://assets.fishersci.com/TFS-Assets/CCG/EU/Sartorius-Stedim-Biotech/brochures/SAR058_EN%20Broch_MD8_Airport.pdf)

  1. Clarify the description of the samples (aerosol and waste water):

Aerosol samples were collected using 3 devices and designated as follows:

  • Airport MD8: samples MD8_CTRL and MD8_tank;
  • Biosampler: samples SKC_PBS_tank;
  • BioSpot-VIVA: samples Vivas_PBS_tank and Vivas_PS_tank.

Clarify the difference between different samples collected using the same device (conditions, sampling site, etc.). On which day were these aerosol samples collected (day 1…..7)?

Clarify the correspondence of aerosol samples and water samples (collection day).

Water samples were designated “Water_1”, “Water_2”, “Water_3”, “Water_4”, “Water_5”, “Water_6”, and “Water_7”, “Water_24h_1”, “Water_24h_2”, “Water_24h_3”, and Water_24h_4”.

Clarify designations “Water_24h_1….4”. On which day were these samples collected (day 1…..7)?

Re: This information is now added in the text (from line 169 to 181, and from 266 to 272), as well as a new table (Table 3) summarizing the sampling design and the characteristics of the samples.

  1. Line 282. A total of 70 087 trimmed reads and 75 OTUs were obtained from water samples, and 18 598 of trimmed reads and 68 OTUs. Provide the data on the number of reads used for each sample (in the text / or the table).

Re: A table containing the number of reads for each sample is now added in the Supplementary Information as Supplementary Table 2. The numbers reported were slightly varied after removing one data point as requested by Reviewer 1.

  1. Figure 1 and results description. The results for the sample “Vivas_PS_tank” significantly differ from those obtained for other aerosol samples. Describe sample “Vivas_PS_tank” and its differences from other aerosol samples. 

Re: A brief description of the “Vivas_PS_tank” sample referring to Figure 1 is now added in the text (line 339).

  1. Figure 1 shows only main genera revealed in the samples. Add the table (for example, in supplementary material) with the list of genera detected in the samples. 

Re: A table with the list of OTUs found in each sample is now added in the Supplementary Information.

  1. The Discussion provides the analysis of the results obtained in the present work and comparison of the devices used.

Compare the results obtained in your work with those obtained in other works to clarify novelty and significance of your work.

Re: As you and another reviewer suggested, we added at the end of the Introduction section (line 130-150) and in the Discussion and Conclusion sections some information about other bioaerosol studies conducted in wastewater treatment plants. However, as we now highlighted in the text, it is difficult to really compare our results with other studies since most of the studies found in literature are based on traditional culture-based methods. Furthermore, the majority of the bioaerosol studies in the WWTP environment do not use the same sampling methods and devices employed in the present work. As in answer to question 4 of reviewer 2 “A comparative table of other WWTPs-based published studies on microbial aerosols in now added in the Supplementary Information as Supplementary table 1 (Table SI1), focusing on studies on non-culturable and culturable bacteria characterization by 16S rRNA gene metabarcoding, on bioaerosol from WWTPs: quantitation is provided by heterogeneous parameters (as CFU/m3 for culturable bacteria, ASV or OTU per sample, gene copies per ng of DNA) not allowing direct comparison among studies.  Qualitative analysis has also to be contextualized: in general, predominant genera are identified, but studies differs in experimental designs, covering single or several seasons, looking for ratios among culturable and non-culturable bacteria, or focusing on pathogens. Pieces of information result hardly comparable”.

The proposed study has the strong point that one of the samplers used (Biospot ViVAS based on condensation growth tubes) provides excellent / undisputed collection efficiencies on very small aerosol particles, without provoking significant stress on biological structure, candidating this as a reference sampling instrument also for bacterial biodiversity studies, and this is reflected by the first experimental data here presented.

Round 2

Reviewer 1 Report

Comments and Suggestions for Authors

The work entitled “Bioaerosol sampling devices and pretreatment for bacterial characterization: theoretical differences and a field experience in a wastewater treatment plant”, authors are Anastasia Serena Gaetano, Sabrina Semeraro, Enrico Greco, Andrea Cain, Maria Grazia Perrone, Alberto Pallavicini, Samuele Greco, Sabina Licen, Stefano Fornasaro and Pierluigi Barbieri, is devoted to an interesting, mysterious and important topic of origin and diversity of microorganisms in the air. The bacterial diversity in aerosols formed near a wastewater treating plant was compared with that of wastewater itself. Some microbial groups, dangerous for workers, such as Enterobacter, Klebsiella and Pseudomonas were found in the air close to WWTP. Bacterial diversity was not strong but correlated with diversity in wastewater. The most appropriate method for cell concentration from air was selected. It was based on application of a BioSpot-VIVAS bioaerosol sampler (the key element of this sampler was a condensation growth tube) and tDNA/RNA shield™ as a collection medium for protection of nucleic acids.

Authors have responded all comments and made all necessary corrections after the first round of revision. Experimental conditions were given in details, especially about normalization between samples and about equity of initial conditions. For future experiments in the area of air microflora studies, it is recommended to repeat air sampling in some time intervals at least three times, i.e perform three independent experiments (it will produce real replicates – see comment 1).

Overall recommendation

Accept

Author Response

We would like to thank Reviewer1 for the indications that helped to improve the quality of the paper. 

Reviewer 2 Report

Comments and Suggestions for Authors

In the rebuttal, authors' response to most of the comments looks fine. 

Comments on the Quality of English Language

This manuscript needs checking of minor typographical related errors. 

Author Response

We would like to thank the reviewer again for the precious advice. The manuscript has now been checked for typographical errors.

Reviewer 3 Report

Comments and Suggestions for Authors

In general, authors provide responses to the reviewer commentaries, therefore the article may be accepted after minor revisions:

1. The information on sampling placed in Figure 1 caption should be placed to Table 3 (information on PBS and PS indexes in sample designations).

2. Line 272. Explain the term “composite process of sampling”

Author Response

We would like to thank the reviewer again for the precious advice. Here follow our responses to the points raised:

1. The information on sampling placed in Figure 1 caption should be placed to Table 3 (information on PBS and PS indexes in sample designations).

Re: Table 3 is now modified in order to further describe the samples. In particular, the column “support medium” is added.

2. Line 272. Explain the term “composite process of sampling”

Re: A description of the composite sampling is now added to the text.